# A daily sunshine duration (SD) dataset in China from Himawari AHI imagery (2016-2023)

Zhanhao Zhang[1,2], Shibo Fang[1], Jiahao Han[1]

[1]State Key Laboratory of Severe Weather, Chinese Academy of Meteorological Sciences, Beijing 100081, China

[2]College of Earth and Planetary Sciences, University of Chinese Academy of Sciences, Beijing, 100049, China

*Correspondence to*: Shibo Fang (sbfang0110@163.com)

**Abstract.** Monitoring global radiation resources relies on sunshine duration (SD) as an important indicator; however, research examining high-resolution SD data is scarce. This study established a daily 5 km SD dataset in China from 2016 to 2023 using Himawari's Advanced Himawari Imager (AHI) Level 3 shortwave radiation fitted with the Ångström–Prescott model based on a time series. We used ground-measured SD at 2,380 Chinese meteorological administration stations to verify the SD dataset accuracy. The results of the testing set indicate that the average correlation coefficient between the SD from the estimation and ground measurements was 0.88. Additionally, we investigated the effects of wind speed, vapour pressure, precipitation, aerosol optical depth, and cloud capacity on the estimation performance of SD and found that temperature had the greatest effect. We also found that cloud capacity that was both too low and too high, and wind speed that was too high affected SD estimation on an average annual scale. These high-resolution SD data provide important support for accurate radiation resource assessments in China. The SD dataset is freely accessible at https://doi.org/10.57760/sciencedb.10276 (Zhang et al., 2024).

## 1. Introduction

Solar radiation is a major driver of photosynthesis and evapotranspiration, plays an indispensable role in regulating temperature and supporting agricultural production, and affects photovoltaic power generation, making it critical to ecosystems and productive human life (Yu et al., 2022; Feng et al., 2021). Satellite remote sensing is an effective method for monitoring and tracking solar radiation, particularly using geostationary satellites that can monitor solar radiation levels in the same target area several times

a day. However, solar radiation inverted by satellite sensors based on reflectance information from land
surfaces is highly susceptible to atmospheric inverted radiation from clouds and aerosols, which must be
corrected using ground measurement radiation stations.
There are limited numbers of radiation observation stations in China (<200 stations in mainland
China) and other parts of the world (Liang et al., 2006; Zhang et al., 2015). This is because of the
expensive upkeep of terrestrial radiation-measuring devices (Zhang et al., 2017; Chukwujindu et al.,
2017), as well as the lack of widely used empirical physical models for satellite-ground radiation
correction, making precise tracking of high spatiotemporal solar radiation over time difficult. Sunshine
duration (SD) is a readily available and cost-effective indicator for monitoring global radiation resources,
and its variability is determined by a combination of regional factors as well as solar constants, cloud
cover, water vapour, and atmospheric pollutants. The SD measured from regular meteorological
observations has the advantages of being over a long period and having good continuity, high spatial
density (>2000 stations in mainland China), and reliability, and is considered the best alternative to solar
radiation (Xia, 2010). SD is a key parameter in solar power potential forecasting (Baumgartner et al.,
2018; Liu et al., 2022); for example, a new SD conversion method based on predicted temperature and
weather type data for daily scale solar radiation prediction was proposed by Qin et al. (2023). Climate
change assessment and agricultural production also need to consider the impact of changes in SD
(Ghanghermeh et al., 2022). Marsz et al. (2021) suggests that long-term variations in SD in Central
Europe are related to changes in the annual frequency of macro-types of circulation in the mid-
troposphere as well as changes in the surface composition of the thermohaline circulation in the North
Atlantic. In addition, some researchers have found that changes in SD also affect the probability of human
diseases (Chang et al., 2022; Gu et al., 2019). Liu et al. (2023) observed that insufficient SD (<5.3 h) was
associated with increased hospitalisation for schizophrenia.
Accurate SD inversion is an important reference for agricultural production, solar resource
utilisation, and global climate change analysis. Studies on SD have mostly been based on limited ground
stations (Vivar et al., 2014; Fan et al., 2018; Yao et al., 2018), while SD is affected by atmospheric
conditions, and it is difficult for a single station to represent this over a large area. Therefore, there is a
considerable need for high-resolution SD data based on satellite remote sensing for studies on solar
radiation. The Advanced Himawari Imager (AHI) instrument, carried onboard the new generation of
geostationary satellites—Himawari-8 and -9—has been widely used to estimate radiation indicators at

different time scales for their shortwave radiation products (Damiani et al, 2018; Hou et al., 2020; Letu et al., 2020; Tana et al., 2023). The Ångström–Prescott model (Ångström, 1924) is the most dominant and widely used model based on SD and solar radiation, and its quadratic and cubic forms have been improved and applied to different meteorological conditions (Rietveld, 1978; Bahel et al., 1987; Chen et al., 2004; Wu et al., 2007; Liu et al., 2012; Ampratwum et al., 1999; Elagib et al 2000). Therefore, based on the advantages of the high spatiotemporal and temporal resolution of the AHI and the existing widely used empirical relationship model between solar radiation and SD, we can use the radiation products of the AHI to validate SD data from high-density regular meteorological observation stations in China to estimate the gridded SD data.

In this study, we generated a daily SD dataset for China at a spatial resolution of 5 km using AHI L3 shortwave radiation data from 2016 to 2023 fitted with the Ångström–Prescott model on different days of the year (DOY). We validated and assessed the accuracy of daily SD data using ground-measured SD and other meteorological data (wind speed, vapour pressure (VAP), cloud capacity, and precipitation) from 2,380 Chinese meteorological administration (CMA) stations, as well as the aerosol optical depth (AOD) from MODIS.

## 2. Data and method

### 2.1 Remote sensing data

The geostationary meteorological satellite Himawari was launched on 7 October 2014 by the Japan Meteorological Agency in Tane Ashima, Japan, with its hypocentre located at 0.0° N and 140.7° E, ~35,800 km above the land surface. Compared to other geostationary satellites, the AHI exhibits superior temporal and spatial resolution, reflection band sensitivity, and accuracy (Zhang et al., 2016). The AHI from Himawari-8 and -9 has 16 spectral channels covering the visible to infrared range, with wavelengths ranging from 0.47 to 13.3 μm, providing a wealth of spectral information (Bessho et al., 2016; Kim et al., 2018; Yu et al., 2019). The temporal and spatial resolutions of the land surface products provided by the AHI are 10 min and 5 km, respectively, which are important for understanding spatiotemporal variations on short time scales (Sawada et al., 2019).

Here, AHI Level 3 hourly shortwave radiation (5 km resolution) data from 1 January 2016 to 31 December 2023 were used for SD dataset construction, calculated by plane-parallel theory and considered the top of atmosphere radiation as the difference between the 300-3000 nm solar shortwave

band and reflected solar radiation by the atmosphere/land surface (Frouin et al., 2007). This approach assumes that the effects of clouds and a clear atmosphere can be decoupled, which has proven to be effective (Dedieu et al., 1987; Frouin and Rachel, 1995). If a one-hour interval was absent from the imagery, linear interpolation was conducted on each pixel of the missing imagery based on the time series. When imagery was absent for a period exceeding one hour, the day in question was excluded. We calculated the daily average shortwave radiation in China based on China Standard Time using hourly AHI shortwave radiation data.

MCD19A2 is a MODIS Terra and Aqua combined multi-angle implementation of atmospheric correction (MAIAC) land AOD-gridded Level 2 product. It is produced daily at a 1 km pixel resolution, which is corrected for atmospheric gases and aerosols using a new MAIAC algorithm that is based on a time series analysis and a combination of pixel- and image-based processing (Lyapustin et al., 2022). Here, AOD at 550 nm in MCD19A2 from 2016 to 2023 was collected using Google Earth Engine (Gorelick et al., 2017).

**2.2 Ground Measurements data**

Ground measurements in the CMA from 1 January 2016 to 31 December 2023 were used to perform the SD estimation. The spatial coverage of Himawari is 2,380 CMA automatic meteorological stations in China. The CMA performs quality control of the data, including spatiotemporal consistency checks, manual corrections, and adjustments, before releasing the meteorological data (Moradi, 2009; Tang et al., 2010). Although the quality of ground-based measurements should be controlled before acquisition, there is still a need for a more stringent check on the data quality based on the daily meteorological data reconstruction method from the CMA (Zhang et al., 2015). Figure 1 shows the spatial distribution of the 2,380 meteorological stations. Here, daily SD, vapour pressure, temperature, wind speed, cloud capacity, and precipitation from the CMA automatic meteorological stations were used to fit and validate the grid dataset as well as to analyse the factors influencing the estimated performance. In this study, March–May was classified as spring, June–August as summer, September–November as autumn, and December–February as winter.

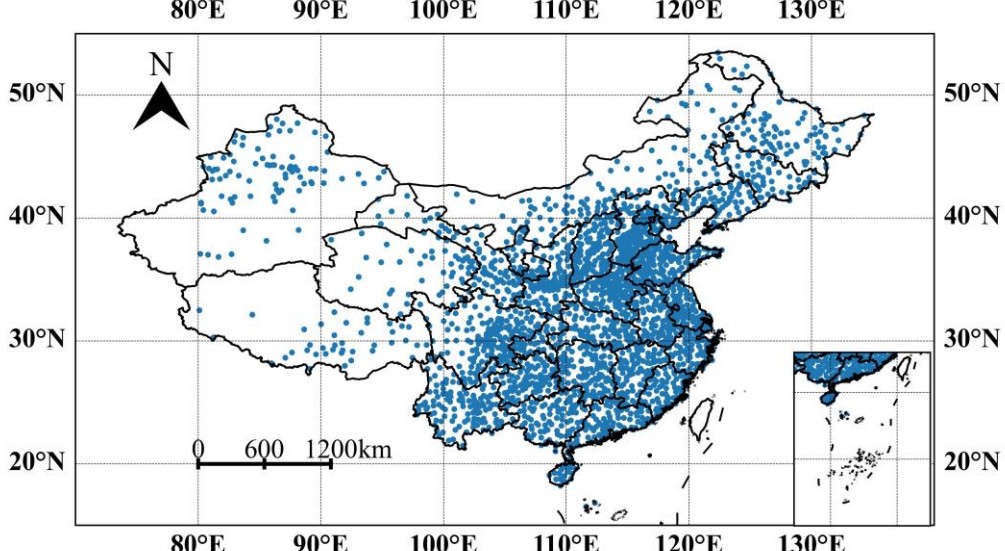

Figure 1. Spatial distribution of the 2,380 automatic meteorological stations of the China Meteorological Administration.

## 2.3 Model overview

The Ångström–Prescott model is an empirical model based on the relationship between SD and solar radiation and is widely used in meteorology and agricultural science. The model was proposed by Ångström based on the total solar radiation on clear days and was improved by Prescott based on astronomical radiation (Ångström, 1924) using the following equations:

$$R_s=(a+b\frac{n}{N})R_a \tag{1}$$

where $R_s$ is the total solar radiation reaching the surface; $R_a$ is the astronomical radiation; a and b are empirical coefficients; n is the actual SD; and N is the maximum SD available. $R_a$ and N counts were calculated according to Liu et al. (2009), as follows:

$$R_a=37.6d_r(\omega_s\sin\varphi\sin\delta+\cos\varphi\cos\delta\sin\omega_s) \tag{2}$$

$$d_r=1+0.033\cos(\frac{2\pi}{365}DOY) \tag{3}$$

$$\delta=0.4093\sin(\frac{2\pi}{365}DOY-1.39) \tag{4}$$

$$\omega_s = \arccos(-\tan\varphi\tan\delta) \tag{5}$$

$$N = \frac{24}{\pi}\omega_s \tag{6}$$

where $d_r$ is the eccentricity of the Earth's orbit around the Sun, $\omega_s$ is the angle at sunset, $\varphi$ is the latitude, $\delta$ is the inclination angle of the sun, and DOY is the days of a year. We considered the AHI Level 3 hourly shortwave radiation as $R_s$ in this model, SD of ground-based observations as a validation of n, and parameters a and b of the Ångström–Prescott model were fitted using the least-squares method.

**2.4 Validation**

We divided the original data into a training set ($>5 \times 10^6$ grid cells during 2017–2022) and a testing set (2016 and 2023 were used as there was a widespread transition from manual to automatic SD recorders in 2019 or station relocations (He et al., 2024)). To identify the best Ångström–Prescott model and corresponding parameters, its performance on the training set (2017–2022) was evaluated using a 100-fold cross-validation (CV) approach and DOY-based CV strategy. In each iteration of each DOY, 99 folds were used as the training set, and the remaining folds were used as the validation set. The training and validation processes were repeated 100 times to obtain the best model parameters a and b for each DOY. In addition, the 2016 and 2023 ground-based SD data were used as test data to evaluate the generalisation capability of the best model parameters, a and b, at each DOY (Figure 2). Pearson's correlation coefficient (R) and root-mean-square error (RMSE) were calculated to evaluate the performance of the model.

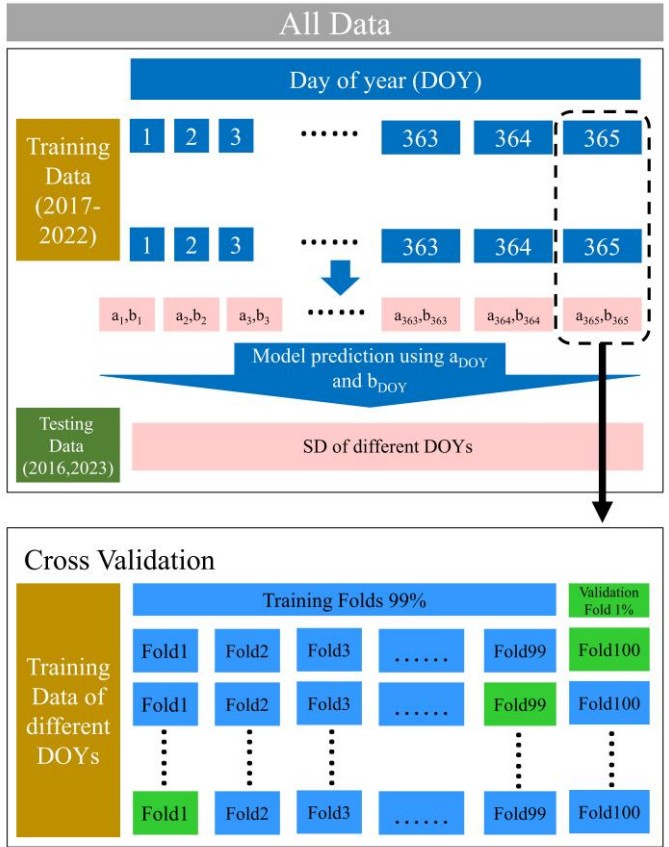


Figure2. Detailed process of model cross-validation and testing.

**2.5 Methods of spatiotemporal variation analysis**

Empirical orthogonal function (EOF) decomposition is an important technique used to investigate

geographical and temporal fluctuations in meteorological characteristics (Zhou et al., 2021). The variable
field can be decomposed into two parts: a spatial function that remains constant across time and a
temporal function that changes exclusively with time; thus, the primary spatial and temporal variations
are evident in the area with a notable contribution to the variance. The spatial function component
comprises several mutually independent and orthogonal spatial modes, which are also considered
eigenvectors. The temporal function consists of the projection of spatial modes in time, which is
represented by time coefficients. We used EOF to analyse the spatiotemporal variations of the established
SD dataset in China. Then, the original variable field information and spatial coefficients were
concentrated in the first few modes.

**3. Results**
**3.1 Evaluation of the training data**

Figure 3 shows the estimation results of the CV sampling method for all DOYs in the training set (N = 68806). An R value of 0.9695 was obtained for the entire training set, with a corresponding RMSE value of 1.2 h. The measured and inverted SD converged to the 1:1 trend line; however, an overestimation occurred in the dense region at ~10 h. Figure 4 shows the inverse performance for different seasons in the training set. The SD was significantly higher in spring and summer than in autumn and winter and was more concentrated in the 0 and 10 h regions in winter. In spring the highest R and RMSE values were 0.9747 and 1.18 h, respectively, while in winter the lowest RMSE value was 1.13 h (Figure 4). However, in summer the highest RMSE value was 1.3 h, and the estimation in summer performed the worst when the measured SD was 0 h. The measured and inverted SD in spring mostly converged to the 1:1 trendline, while overestimation occurred in the dense region around 10 h in winter.

Figure 5 shows the optimal Ångström–Prescott model parameters a and b for different DOYs. The parameter a has an upward parabolic trend with DOY, with local maximum and minimum values of 0.22 at DOY = 306 and 0.13 at DOY = 351, respectively. Parameter b showed a significant "W"-shaped variation with DOY, with a local maximum value of 0.74 at DOY = 146 and two local minimum values of 0.66 and 0.63 at DOY = 99 and 351. In general, parameters a and b of the Ångström–Prescott model are characterised by more pronounced seasonal variations. Figure 6 shows the variation in the training set evaluation indicators (R and RMSE) with DOY. More than half of the DOYs had R values greater than the overall R value (Figure 3), but there were still 134 days with R values <0.97 and a minimum value of 0.94 at DOY = 193. Meanwhile more than half of the DOYs have RMSE values less than the overall RMSE values in Figure 3, but there were still 157 days with R values <1.2 h, and again there was a maximum value of 2.1 h for RMSE at DOY = 193. The evaluation indicator for the training set was not characterised by significant seasonal variations.

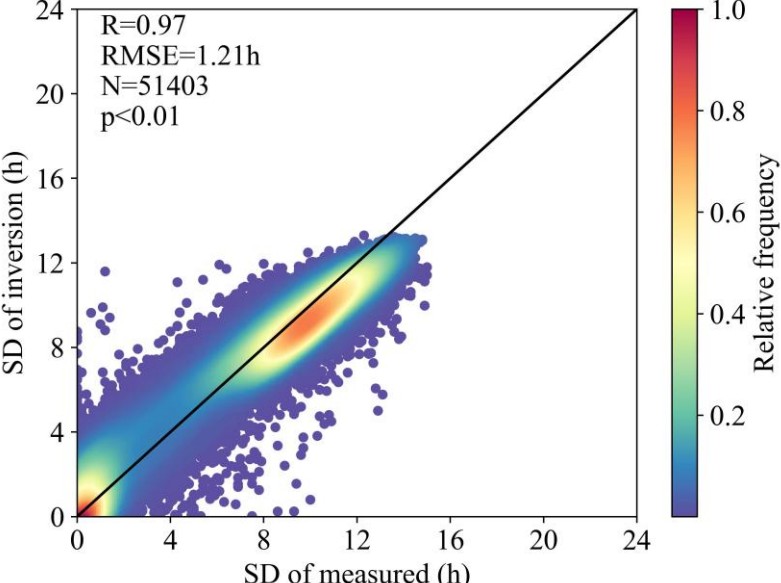


Figure 3. Estimation results of the CV sampling method in the training set.


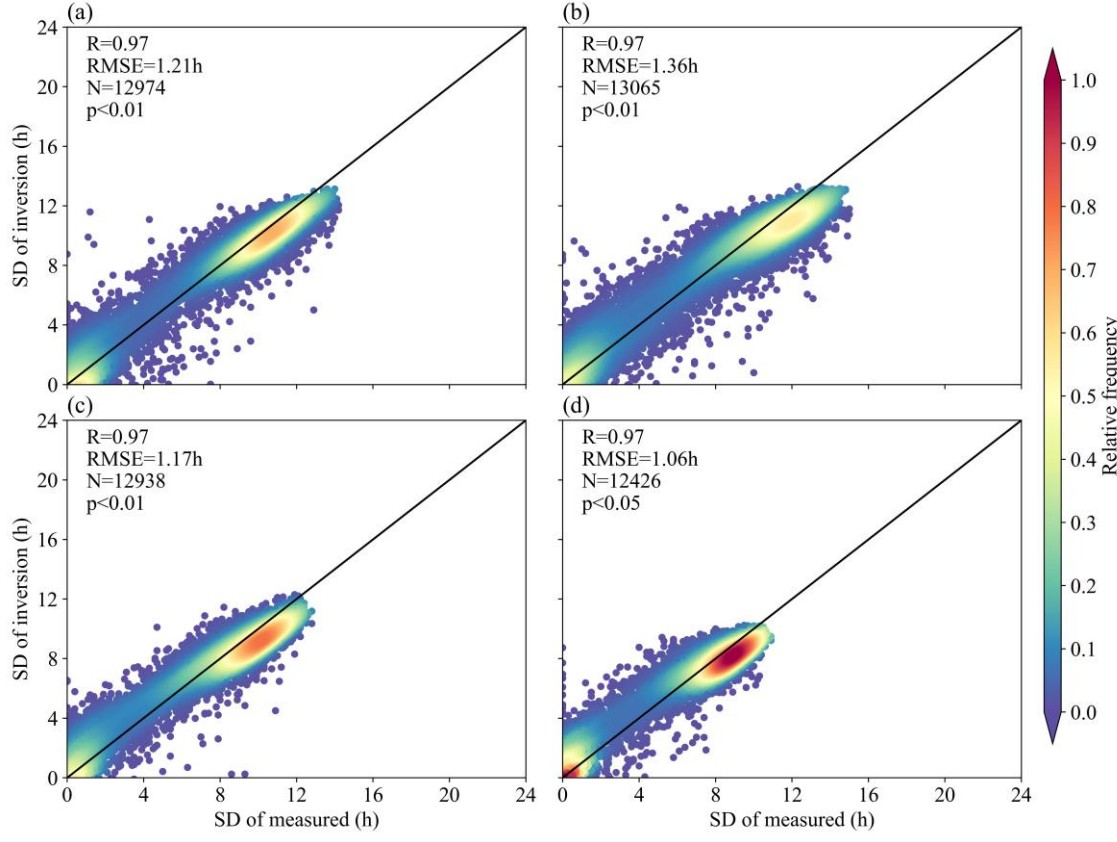


Figure 4. Estimation results of the CV sampling method in the training set from different seasons: (a)

spring, (b) summer, (c) autumn, (d) winter.

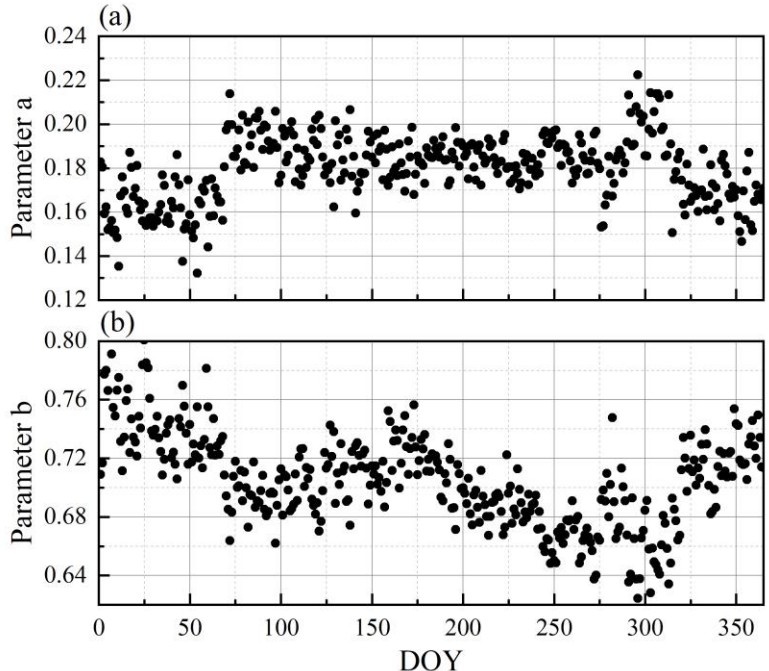


Figure 5. The a and b coefficients of the Ångström–Prescott model for different DOYs.

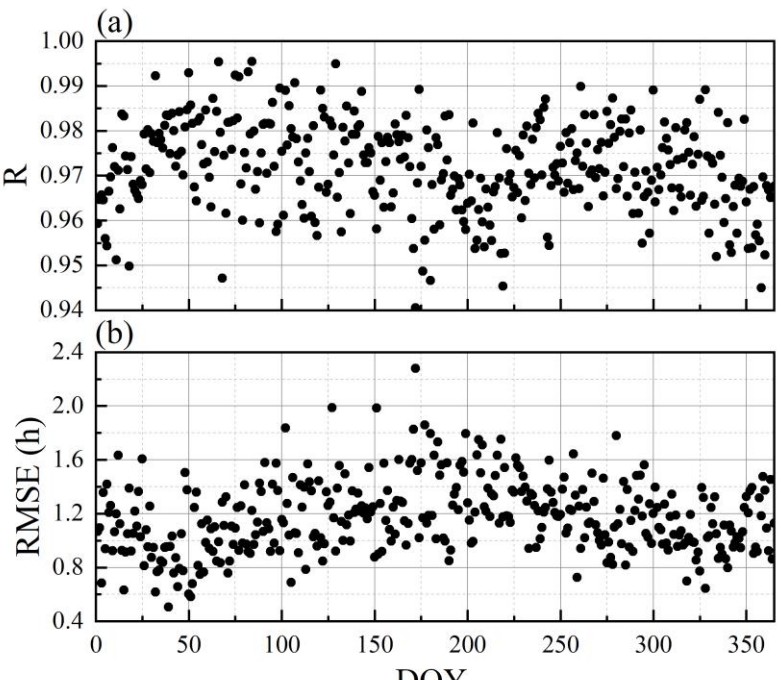


Figure 6. The correlation coefficients (R) (a) and RMSE (b) of the CV sampling method in the training

set for different DOYs.


**3.2 Evaluation of the testing data**

The different evaluation indicators for the test sets (2016 and 2023) are shown in Figure 7. Figure

7(a) shows the R of 2016 and 2023, with the trends in these two years essentially identical with an "M"
shape. The average R value for 2016 was 0.88, which is generally consistent with that for 2023. The
minimum R value of 0.52 in 2023 (DOY = 361) was lower than that of 0.60 in 2016 (DOY = 21), but
both occurred during winter. The trend of RMSE values for 2016 and 2023 is opposite to the R value,
with the maximum and minimum RMSE values occurring in 2023 at 2.77 (DOY = 355) and 1.19 (DOY
= 106), respectively. Figures 7(c) and (d) show the estimated performances of 0 SD (no sunshine for the
entire day) for the CMA meteorological stations in 2016 and 2023. Figure 7(c) shows the estimated mean
values of 0 SD for different DOYs in 2016 and 2023, where the mean value in 2023 (0.49 h) is smaller
than that in 2016 (0.75 h), with the maximum and minimum mean values still occurring in 2023 at 3.42
(DOY = 211) and -0.75 (DOY = 134), respectively. Figure 7(d) shows the number of estimated SD <0
for different DOYs in 2016 and 2023. There were more average daily estimated SDs of <0 in 2016 than
in 2023 at 267/day, with the lowest value occurring in 2016 at 997 (DOY = 294). The bias in the 0 SD
estimation is linked to the over- and under-representation of its number. Changing all estimated SDs from
<0 to 0 resulted in an improvement in their estimated performance (Figure 8), with 2016 showing a
greater improvement than 2023 with DOY = 285.

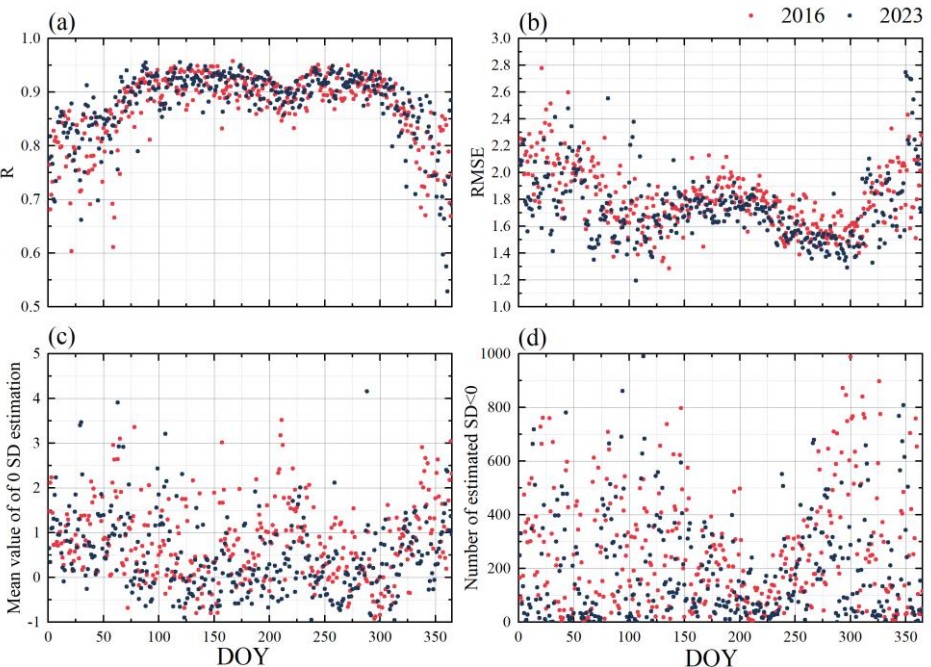


Figure 7. Estimated performance in the testing set.

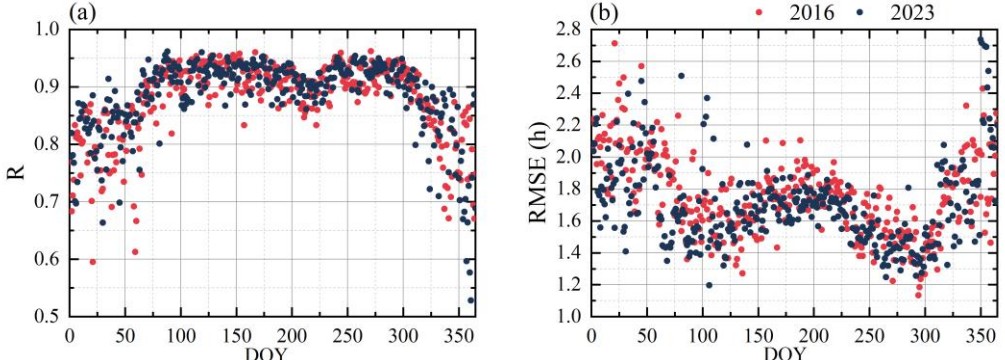


Figure 8. Estimated performance by changing all estimated SDs <0 to 0 in the testing set.


**3.3 Effect of different environmental factors on SD estimation**

Figure 9 shows the effects of the national daily average VAP, precipitation, and temperature (based

on CMA meteorological stations) on the R values in Figure 8. The R values were exponentially related
to both VAP and precipitation, and VAP had a greater effect on R than precipitation. Meanwhile, the
estimated performance in 2016 was more affected by moisture conditions. Temperature had the greatest
impact on R, with 2023 being affected to a greater extent than in 2016 (Figure 9 (e, f)). The influences
on SD estimation are discussed by distinguishing the different seasons (Table 1). VAP, precipitation, and
temperature had the greatest influence on R values in autumn and the least in winter. Notably, R in
summer was negatively correlated with VAP and temperature.

Figures 10 and 11 show the annual average SD from the CMA meteorological stations and

Himawari estimations in 2016 and 2023, respectively, along with the annual average AOD, wind speed,
and cloud capacity. On an annual scale, ground-measured and estimated SD are more consistent in eastern
and northern China, while both years have higher estimates in eastern China and lower estimates in
northwestern and northeastern China. When comparing the impact factors, higher wind speed, and lower
AOD in these areas, both affected the SD estimation. The estimated SD appears to be overestimated
(southern China) at excessively high cloud cover, especially at excessively high low-cloud cover, which
was more pronounced in 2016. In 2023, the total cloud cover was higher, the low-cloud cover was lower,
and the estimation error had poorer feedback on the cloud cover.

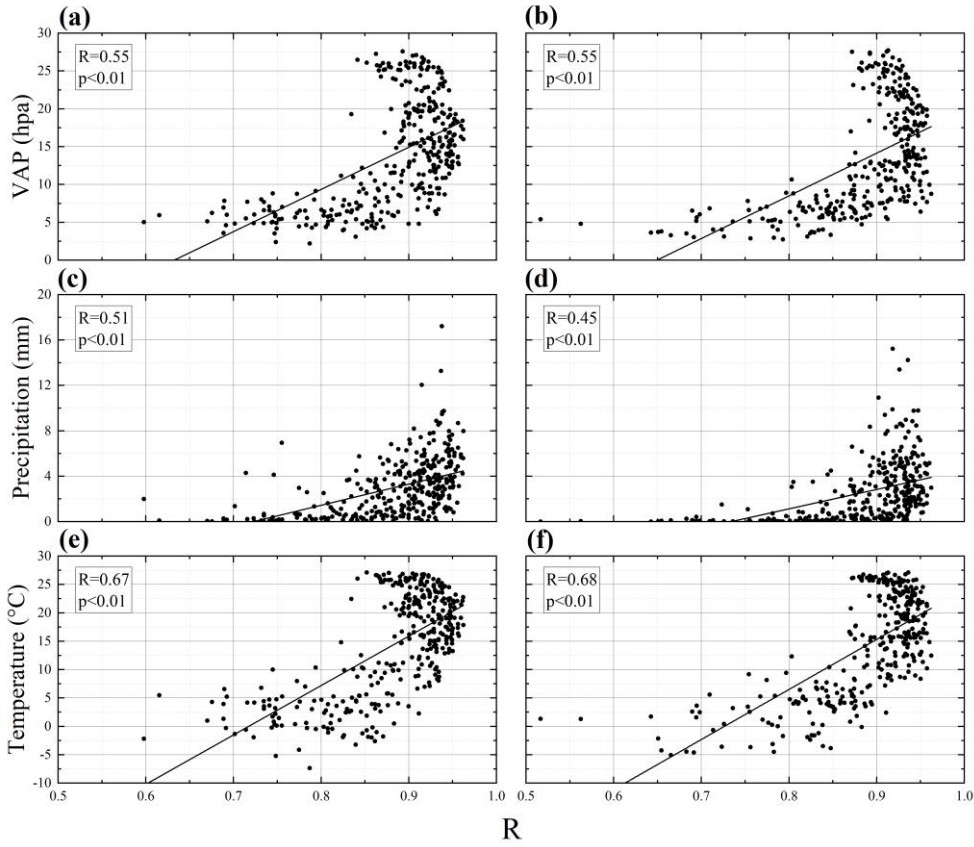


Figure 9. R values and different environmental factors: VAP (a, b), precipitation (c, d), temperature (e,

f). Correlations in 2016 (a, c and e) and 2023 (b, d and f).


**Table 1.** Correlation coefficients between estimated performance and influencing factors in different

seasons (* and ** refer to passing the $p < 0.05$ and $p < 0.01$ significance tests, respectively)

| Time | Influencing Factors | | |
|------|------|------|------|
| | VAP | Precipitation | Temperature |
| Spring | 0.29* | 0.43** | 0.31* |
| Summer | -0.56* | 0.28* | -0.53** |
| Autumn | 0.59** | 0.46** | 0.62** |
| Winter | 0.28* | 0.26** | 0.22** |


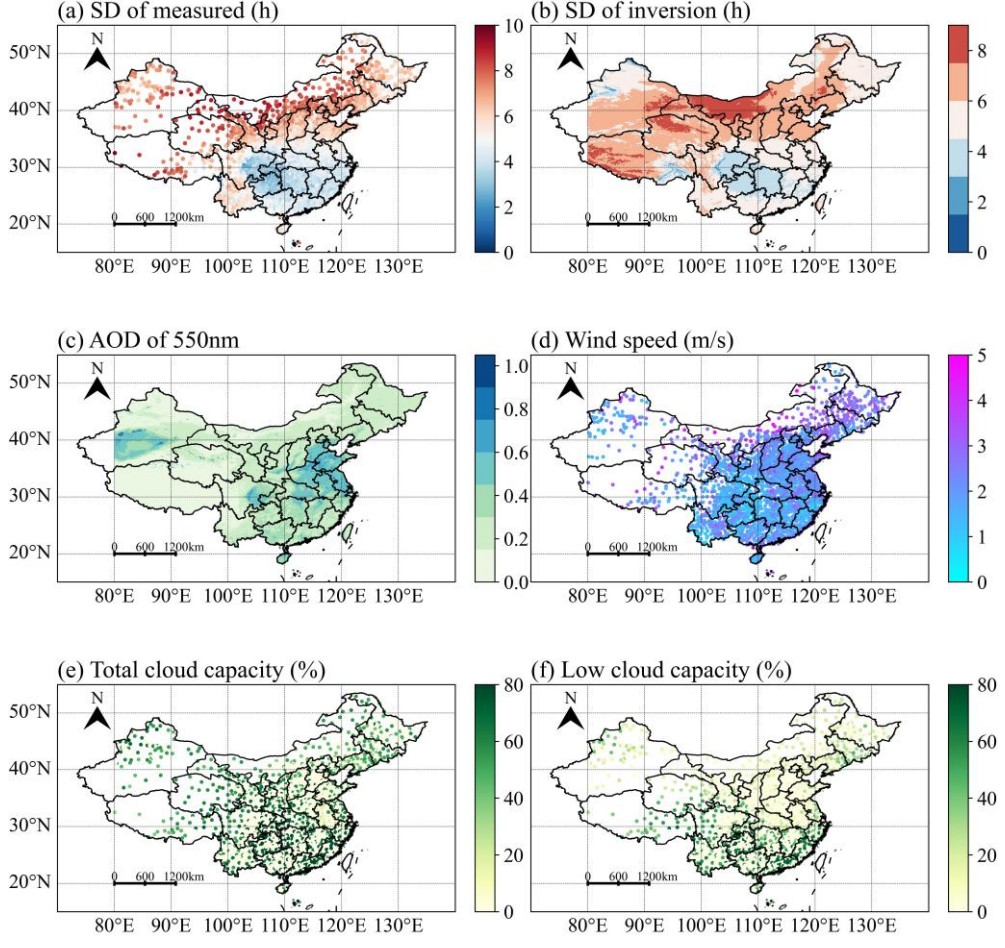


Figure 10. Comparison of annual average ground measurement (a) and Himawari (b) SD in 2023,

giving an annual average AOD of 550 nm (c), wind speed (d), total cloud capacity (e), and low cloud

capacity (f).

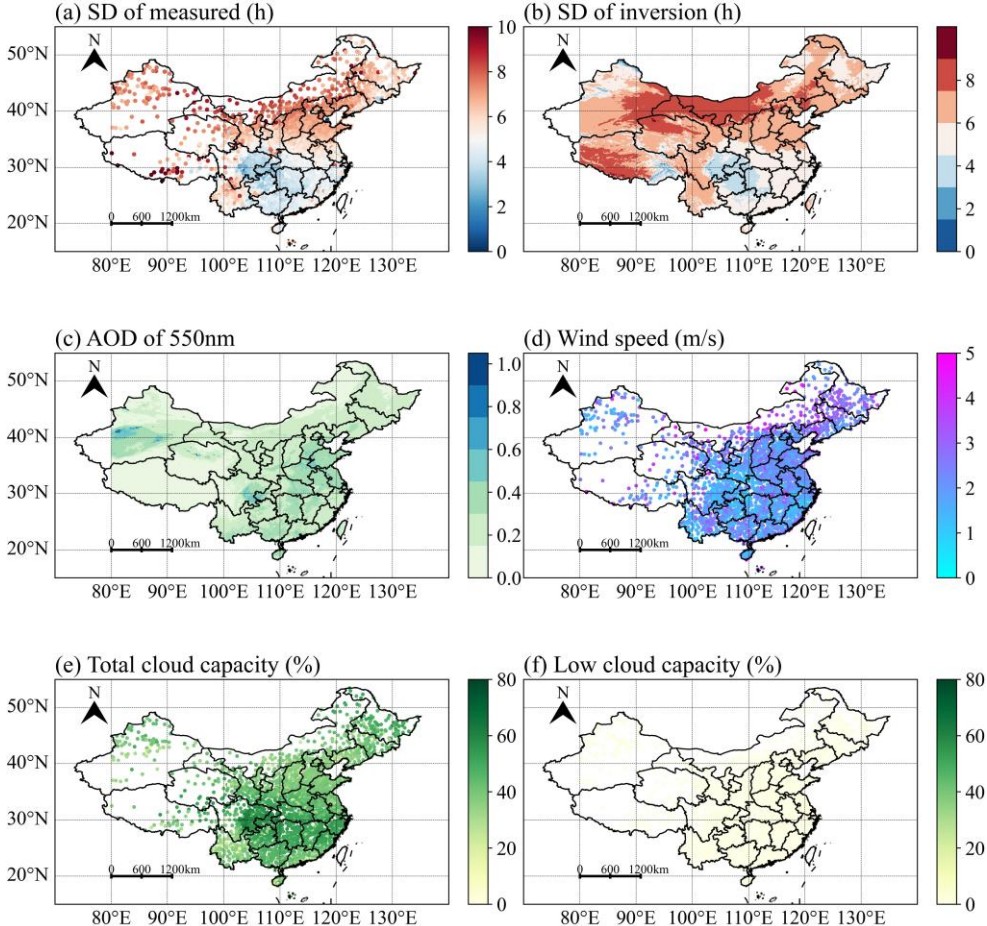


Figure 11. Comparison of annual average ground measurement (a) and Himawari (b) SD in 2023,
giving an annual average AOD of 550 nm (c), wind speed (d), total cloud capacity (e), and low cloud
capacity (f).

**3.4 Effect of different environmental factors on SD estimation**

The EOF analysis of the mean annual SD grid data in China from 2015 to 2023, and the spatial

variance contribution rate of the eigenvectors in the first three EOF modes are shown in Figure 12, where
the explained variance of each mode is 30.44, 23.47, and 19.0 %, respectively, with a cumulative variance
contribution of ~72.91 %. The variance contribution rate of the Mode 1 eigenvectors in Figure 12(a)
surpassed that of the other models, making it the predominant spatial distribution in China. Mode 1
decreases from western to eastern China, and northwest China exhibits extremely low values; however,
there are exceptions in Yunnan Province. Mode 2 (Figure 12(b)) exhibits a dipolar distribution,
decreasing from southern to northeastern China, and mode 3 shows a tri-pole distribution, decreasing
from central China to the sides. Generally, it can be concluded that SD decreases from western to northern

China. Figure 12(d, e, f) shows the time coefficients of SD from the first three models in China, where the SD time coefficients of Mode 1 (Figure 12(d)) show an increasing trend from 2016 to 2023, with the minimum time coefficient in 2019 and maximum time coefficient in 2021. The SD time coefficients of Modes 2 and 3 exhibit a decreasing trend, and both are positive in 2016 and negative in 2019 (Figure 12(e, f)).

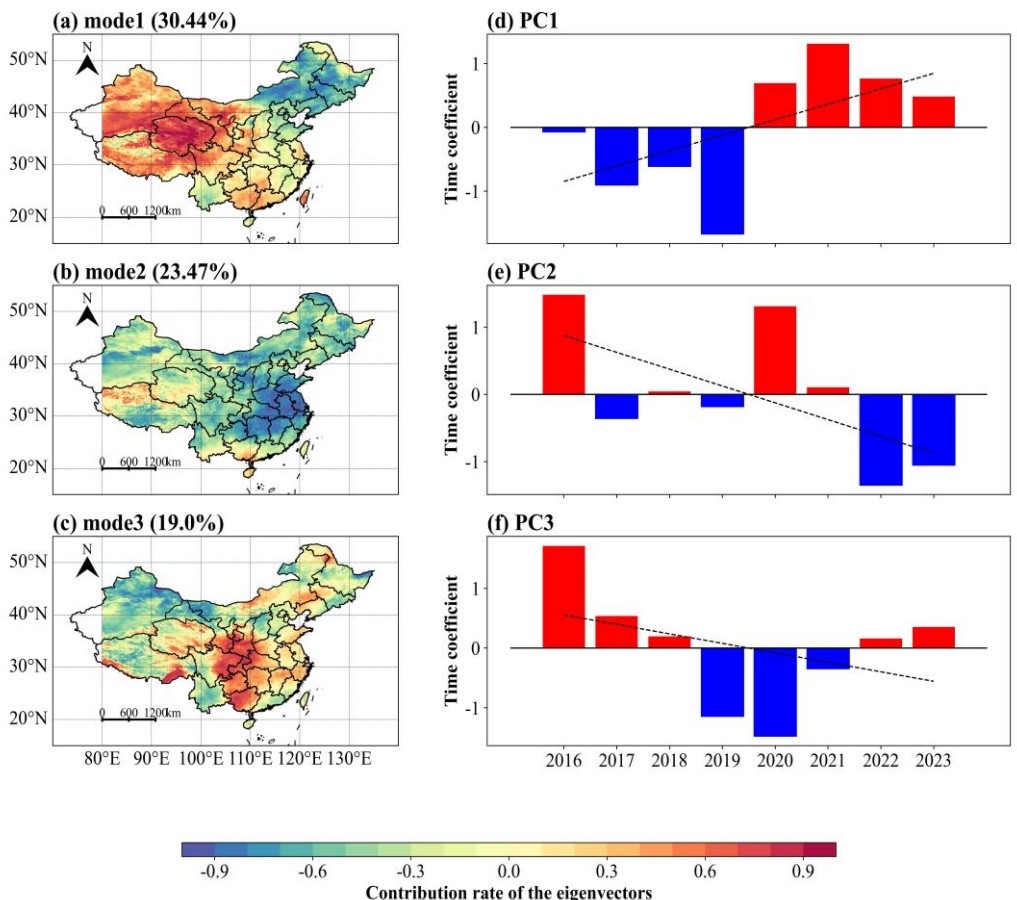

Figures 12. Distribution of eigenvector contribution rates (a–c) and time coefficients (d–f) for the first three modes of SD.

## 4. Discussion

There is no explicit remote sensing inversion model for SD as its observation is founded upon the accumulation of radiation. Consequently, SD datasets were constructed through spatial interpolation, which results in the absence of SD datasets that are released with high spatiotemporal resolution. Here, a 5 km resolution SD dataset in China from 2016 to 2023 was established based on a time series using Himawari imagery fitted with an Ångström–Prescott model, an approach not done before by previous

studies.

The time series based on the Ångström–Prescott model was used to invert the SD in China, setting

the coefficients of a and b to fixed values for the entire region at different DOYs, while the suggested
coefficients in this study are not comparable with the calibrated coefficients for other regions. Previous
studies on the Ångström–Prescott model have confirmed that it is a reliable tool for estimating solar
energy in practical applications with no marked dependence on latitude (Paulescu et al., 2016).
Additionally, the accuracy of the model has a strong dependence on the season (Liu et al., 2023), and
according to the results here (Figs. 4–8), the cause of this can be attributed to differences in the day and
night lengths in different seasons. This work not only provides a more accurate evaluation standard for
the level of radiation received on the ground but also provides better support for radiation estimation in
the future. More conventional meteorological stations will be established in the future to validate and
improve the Ångström–Prescott model based on a time series. It is noteworthy that the number of
meteorological observation stations in southwestern China (especially in the Tibetan Plateau Region) is
small and distributed unevenly, and the snow in the plateau notably affects interpretation of the
reflectance data from the Himawari imagery. Thus, we will consider the input of the land cover
characteristics as the climatological data in following studies to improve this poor performance.

It is worth noting that there is a bias in the validation of the training and test data, where there is an

overestimation at 0 SD (Figure 3), which may be the strong light in most of the area under a DOY, leading
to the Ångström–Prescott model's larger parameters and overestimation of a very small portion of the
image elements that contain aerosols, clouds, and even precipitation. In addition, in the test data the
estimated SD was <0 (Figure 7(c, d)), because the thicker clouds, atmospheric aerosols, and water vapour
in the majority of the area on that day did not have much effect on the ground-based SD instrument (the
atmospheric longwave radiation contained in the direct radiation was not affected) but had a significant
effect on the AHI shortwave radiation data, resulting in an SD of <0. After changing the image elements
with SDs <0 to 0, the validation results remained substantial (Figure 8), indicating that this part of the
radiation was essentially less than the threshold for SD observations (120 W/m$^2$). In conclusion, because
our approach is based on a time series, it is unavoidable that we will encounter input data that are not
sensitive to different sky conditions. In the future, the use of relevant physical precipitation models will
be considered to simulate the precipitation process at different times of the day based on radiation data.
This will enable us to estimate the SD, and this aspect of the Ångström–Prescott model will be
subsequently improved.
We found that temperature, moisture conditions, wind speed, and atmospheric pollutants all
influenced the SD estimation, with temperature having the greatest effect on temporal variation and wind
speed having a stronger effect on spatial variation than AOD and cloud capacity. However, we believe
that the effects of these environmental factors are not independent but are the result of interactions (Tang
et al., 2022). In densely populated and economically developed areas (eastern and southern China), where
pollutant levels are higher and increased wind speeds accelerate their dispersion, this regulatory
mechanism is enhanced by increasing pollutant concentrations (O'Dowd et al., 1993; Wang et al., 2014).
An increase or decrease in wind speed affects the rate of diffusion of water vapour and pollutants in the
air, which in turn affects atmospheric transparency and, ultimately, SD estimation. However, the effect
of temperature on SD estimation in this study was not consistent with that in previous studies (Tang et
al., 2022; Feng et al., 2019; Ren et al., 2017), which suggests that the relationship between SD,
temperature, and relative humidity is complex and needs further investigation.
The EOF method analysis of the mean annual SD indicates that it decreases from western to
northeast China, consistent with the results of Tang et al. (2022) and Xiong et al. (2020) and suggesting
that the pattern of industrial development between western and eastern China affects radiation levels to
some extent. The EOF time coefficients show that there has been a certain degree of increase in SD in
recent years, which correlates with the long-term SD analysis by Tang et al. (2022). This trend may be
related to global climate change (Josefsson and Landelius, 2000), because the variation in wind speeds
due to global warming has resulted in decreased cloud dissipation across mainland China (Xiong et al.,
2020). In addition, the decrease in human activity in recent years (Liu et al., 2020) has contributed to the
weakening of the urban rain island effect and aerosols (Glantz et al., 2006), and it appears that the latter
factor is more influential in this study. However, short-term reductions in human activity cannot become
the norm, and sunshine duration is bound to fluctuate due to the acceleration of the hydrological cycle.

**5. Data availability**

The SD dataset is freely accessible at https://doi.org/10.57760/sciencedb.10276 (Zhang et al., 2024).

**6. Conclusion**

We introduced a newly developed high-resolution dataset that provides SD data for China for the
period 2016–2023. We calculated the daily SD using Himawari Level 3 shortwave radiation fitted with

the Ångström–Prescott model based on a time series and used ground-measured SD to evaluate the

estimation performance. The validation of testing data from ground-measured SD gave favourable results,

with R values >0.5 and an average of 0.88 for all days in 2016 and 2023. We also found that temperature

and wind speed dominated the Ångström–Prescott model in estimating SD. A future direction for this

study would be to divide the Chinese regions into suitable areas to independently estimate and synthesise

a more accurate daily SD dataset for China.

**Author contributions.** ZZ and SF designed and organized the paper. ZZ and JH prepared the related

materials and ran the dataset. ZZ evaluated the accuracy of the dataset. All authors discussed the results

and commented on the paper.

**Competing interests.** The contact author has declared that none of the authors has any competing

interests.

**Financial support.** This research was supported by the National Key Research and Development

Program of China (grant no. 2023YFE0122200), the National Nature Sciences Foundation (grant no.

42075193).

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
