# Peer review of "A daily sunshine duration (SD) dataset in China from"

_Earth System Science Data, 2024_

## Author Comment (AC1)

Dear Referee 1, After receiving your comments, we immediately prepared a response, and the following is a point-by-point response to all the questions.

*I have two main questions on this paper. Firstly, why do you focus on sun duration instead of incident solar radiation at land surface? In which sector or subject, sun duration is indispensable not solar radiation? I think this should be highlighted in introduction and also should be discussed in detail in discussion. Secondly, is there any released sun duration dataset? If not, why researchers not establish such dataset? What is the bottleneck in theory and in technology? If yes, please compared your results with other similar datasets.*

Thank you for your valuable question, it was a pleasure to discuss this with you and we will answer it in the following two points:

(1) The incident solar radiation at the land surface (SISR) can accurately invert solar power potential and involved in climate change and agricultural production. The two main ways of inverting SISR, satellite remote sensing and ground measurement, are presently biased. Satellite sensors invert SISR (also known as incident shortwave radiation) from reflectance information on the land surface, which is very susceptible to atmospheric inverse radiation from clouds and aerosols [1, 2], and therefore the current high spatiotemporal resolution and continuous SISR datasets make it impossible to accurately assess the radiological resources of the region. More importantly, the SISR ground measurement sites in the Chinese region are very few (145), much fewer than conventional meteorological stations observed by sunshine duration (SD) (2408), and therefore the large-area SISR verification is difficult **(These have been added in the revised manuscript).**

SD the is widely used for SISR estimation and was presented in FAO56 [3], and subsequently many relationship models with SISR have been proposed for the SD to the maximum SD available (N) ratio [4,5]. Several studies have estimated SD to predict solar power potential [6, 7]. And SD is also an important indicator for assessing climate change trends, and researchers can understand the spatiotemporal variability of solar radiation through high temporal and spatial resolution SD data [8] **(These have been added in the revised manuscript).** In addition, some researchers have found that changes in SD also affect the probability of human diseases [9, 10].

Therefore, we believe that the study on SD estimation is easy to validate, and the SD data are more credible theoretically. Until the radiation observation system in China is well developed, we believe that the study on the estimation of SD is valuable.

(2) As SD observations are founded upon the accumulate of solar radiation, there is no explicit remote sensing inversion model for SD. This study fits the Ångström-Prescott model with SD and Himawari AHI radiation product at all spatial locations to derive the parameters a and b based on different time series. The validation results proved the accuracy and efficiency of our method, which was better than a study that estimated SD from diurnal temperature range [7]. Meanwhile, other studies on spatiotemporal variations in SD are also carried out based on the interpolation of site SD data [8] **(These have been added in the revised manuscript).**

Our SD dataset has received a high number of downloads and visits since its release, and a high number of reprints in mainland China, which proves the value of developing this SD

dataset (http://www.gis5g.com/data/qxsj?id=2563).

Reference:
1. Mei, L., Rozanov, V.V., Vountas, M., & Burrows, J.P. (2018). The retrieval of ice cloud parameters from multi-spectral satellite observations of reflectance using a modified XBAER algorithm. Remote Sensing of Environment.
2. Letu, H., Yang, K., Nakajima, T.Y., Ishimoto, H., Nagao, T.M., Riedi, J.C., Baran, A.J., Ma, R., Wang, T., Shang, H., Khatri, P., Chen, L., Shi, C., & Shi, J. (2020). High-resolution retrieval of cloud microphysical properties and surface solar radiation using Himawari-8/AHI next-generation geostationary satellite. Remote Sensing of Environment, 239, 111583.
3. Allen, R.G., Pereira, L.S., Raes, D., & Smith, M. (1998). Crop evapotranspiration: guidelines for computing crop water requirements.
4. Chen, J., He, L., Yang, H., Ma, M., Chen, Q., Wu, S., & Xiao, Z. (2019). Empirical models for estimating monthly global solar radiation: A most comprehensive review and comparative case study in China. Renewable and Sustainable Energy Reviews.
5. Prieto, J. I., & García, D. (2022). Global solar radiation models: A critical review from the point of view of homogeneity and case study. Renewable and Sustainable Energy Reviews, 155, 111856.
6. Liu, F., Wang, X., Sun, F., & Wang, H. (2022). Correct and remap solar radiation and photovoltaic power in China based on machine learning models. Applied Energy.
7. Qin, S., Liu, Z., Qiu, R., Luo, Y., Wu, J., Zhang, B., Wu, L., & Agathokleous, E. (2023). Short–term global solar radiation forecasting based on an improved method for sunshine duration prediction and public weather forecasts. Applied Energy.
8. Tang, C., Zhu, Y., Wei, Y., Zhao, F., Wu, X., & Tian, X. (2022). Spatiotemporal Characteristics and Influencing Factors of Sunshine Duration in China from 1970 to 2019. Atmosphere.
9. Chang, Z., Chen, Y., Zhao, Y., Fu, J., Liu, Y., Tang, S., Han, Y., & Fan, Z. (2022). Association of sunshine duration with acute myocardial infarction hospital admissions in Beijing, China: A time-series analysis within-summer. The Science of the total environment, 154528.
10. Gu, S., Huang, R., Yang, J., Sun, S., Xu, Y., Zhang, R., Wang, Y., Lu, B., He, T., Wang, A., Bian, G., & Wang, Q. (2019). Exposure-lag-response association between sunlight and schizophrenia in Ningbo, China. Environmental pollution, 247, 285-292.

*1. Why do you choose Himawari AHI? Compared to other shortwave radiation product, what is superiority of this product?*

**Currently, geostationary and polar-orbiting satellite data are widely used for the estimation of radiation indicators. Geostationary satellites have a higher frequency of observations than polar-orbiting satellites and are widely used to generate radiation indicator products (These have been added in the revised manuscript). Since 2015, a new generation of geostationary satellites such as the Himawari and FY series have been launched over the skies, in particular the Himawari AHI, whose high temporal (10 min) and spatial (5 km) compared to other global satellite products (e.g., Modis or Landsat, etc.) has been widely used for the estimation of different radiometric indicators on hourly and daily scales [1-3], and comparing with FY-4, Himawari AHI is superior to the AGRI sensor carried by FY-4 in terms of temporal and spatial resolution, reflection band sensitivity and accuracy [4]. (These have been added in the revised manuscript).**

Reference:

1. Hou, N., Zhang, X., Zhang, W., Wei, Y., Jia, K., Yao, Y., Jiang, B., & Cheng, J. (2020). Estimation of Surface Downward Shortwave Radiation over China from Himawari-8 AHI Data Based on Random Forest. Remote. Sens., 12, 181.

2. Letu, H., Yang, K., Nakajima, T.Y., Ishimoto, H., Nagao, T.M., Riedi, J.C., Baran, A.J., Ma, R., Wang, T., Shang, H., Khatri, P., Chen, L., Shi, C., & Shi, J. (2020). High-resolution retrieval of cloud microphysical properties and surface solar radiation using Himawari-8/AHI next-generation geostationary satellite. Remote Sensing of Environment, 239, 111583.

3. Tana, G., Ri, X., Shi, C., Ma, R., Letu, H., Xu, J., & Shi, J. (2023). Retrieval of cloud microphysical properties from Himawari-8/AHI infrared channels and its application in surface shortwave downward radiation estimation in the sun glint region. Remote Sensing of Environment.

4. Zhang, P., Guo, Q., Chen, B. and Feng, X.: The Chinese Next-Generation Geostationary Meteorological Satellite FY-4 Compared with the Japanese Himawari-8/9 Satellites. Adv. Meteorol. Sci. Technol., (1), 4. https://doi.org/10.3969/j.issn.2095-1973.2016.01.010, 2016. (in chinese)

*2. Please give more information on the algorithm, accuracy and bias of Himawari AHI Level 3 shortwave radiation in 2.1 section.*

**Thank you for this comment. As the AHI merely offers references [1] to the methodology, we have chosen not to include the algorithm, accuracy, and bias of Himawari AHI Level 3 shortwave radiation into the 2.1 section, given the comprehensiveness of the article. The above collection of algorithms contains a greater number of formulas that could potentially impede the reader's access.**

Reference:

1. Frouin, R. and Murakami, H.: Estimating photosynthetically available radiation at the ocean surface from ADEOS-II global imager data. J Oceanogr., 63, 493-503. 2007.

*3. Lines 73-75: Do you want to describe the calculation for surface solar radiation? If yes, I do think it can be obtained by using TOA solar radiation minus solar radiation attenuated amount by atmosphere.*

**We apologize for the confusion, the thinking of the original paper Lines 73-75 [1] is essentially the same as your comment, which we have modified considering your comment in the revised manuscript.**

Reference:

1. Frouin, R. and Murakami, H.: Estimating photosynthetically available radiation at the ocean surface from ADEOS-II global imager data. J Oceanogr., 63, 493-503. 2007.

*4. Line 107: I do think the citation should be (Angstrom, 1924).*

**Thank you very much for your suggestion, we have corrected it.**

*5. Why do you keep four decimals for R but one for RMSE?*

**We are very sorry for the unattractiveness of the article brought about by the different decimal places, and have uniformly changed it to retain 2 decimal places.**

*6. Please add significance test when you showing R.*

Thank you for your comments, we have added Figures 3, 4 and 9 accordingly.

*7. Why do you fix the parameter of a and b across the whole China. Is there no spatial variations in the two parameters?*

Thank you for raising this critical issue, as in the first response, the previous studies on fitting the Ångström-Prescott model with SD and total radiation were often based on different moments at the same spatial location, and the parameters obtained cannot be applied to locations where there are no SD ground measurement sites. This study fits the Ångström-Prescott model with SD and Himawari AHI radiation product at all spatial locations to derive the parameters a and b based on different time series, therefore, the parameters a and b in this study do not vary spatially only temporally.

*8. Figure 3: Why there are many SD estimations with different values (close to 12) when observation is zero?*

Thank you for raising this issue, the reason for this may be that the strong light in almost most of the area under a DOY leads to Ångström-Prescott model larger parameters and over-estimation of a very small portion of the image elements that contain aerosols, clouds and even precipitation, this type of phenomenon accounts for a very small portion of our study, and these will be added to the discussion in the revised manuscript. This aspect of the Ångström-Prescott model will be improved subsequently.

*9. Line 178: Why estimated SD less than 0 happens? You should discuss the reason in detail in discussion.*

Thank you for raising this issue, the reason for this may be that the thicker clouds, atmospheric aerosols and water vapor in majority of the area on that day did not have much effect on the ground-based SD instrument (the atmospheric longwave radiation contained in the direct radiation was not affected), but had a significant effect on the shortwave radiation from Himawari AHI L3, resulting in SD less than 0. After changing the image elements with SD less than 0 to 0, the validation results are still substantial (Figure 8), indicating that this part of the neglected longwave radiation is essentially less than the threshold for SD observations (120 W/m2) (These have been added in the revised manuscript).

*10. Figure 9: I am not sure the correlation coefficients is for spatial pattern or for temporal variation.*

We apologise for the confusion. It is temporal variation, R-values are those in Figure 8, and we have added explanations accordingly in the revised manuscript.

*11. Figure 10: Why only choose 28 September 2016 as a case study? Once you have long-term MODIS AOD data, please try to analyze the relationship between simulation performance and AOD level for whole year or four seasons.*

Thank you very much for your comment, we have changed to an annual average comparison (These have been added in the revised manuscript).

[Figure]

Figure 10. Comparison of annual average ground measurement (a) and Himawari (b) SD in 2016, giving annual average AOD of 550nm (c) and the wind speed (d).

[Figure]

Figure 11. Same as Figure 10, but in 2023.

*12. Once the value you showed have units, please give. Such as in Figure 11.*
We have added in Figure 10 and 11.

*13. At last, you should show the spatial and temporal variations in SD with your datasets.*
Thank you very much for your suggestion, we have added the Empirical orthogonal

**function (EOF) method for spatiotemporal analysis of SD in the revised manuscript.**

"EOF analysis of mean annual SD grid-data in China from 2015-2023, the spatial variance contribution rate of the eigenvectors in the first three EOF modes are shown in Figure 12, where the explained variance of each mode is 30.44%, 23.47% and 19.0%, respectively, with a cumulative variance contribution of about 72.91%. The variance contribution rate of mode 1 eigenvectors in Figure 12a surpasses that of other models, making it the predominant spatial distribution in China. The mode 1 decreases from western to eastern China, the northwest China exhibits extremely low values, but there are exceptions in Yunnan Province. The mode 2 (Figure 12b) exhibits a dipolar-type of distribution decreasing from the southern to northeast China, and the mode 3 shows a tri-pole distribution decreasing from central China to sides. Generally, it can be concluded that the SD decreases from western to northern China. Figure 12def shows the time coefficients of SD from the first three models in China, the SD time coefficients of the mode 1 (Figure 12d) shows an increasing trend from 2016 to 2023, with the minimum time coefficient in 2019 maximum time coefficient in 2021. It can be seen from Figure 12ef that the SD time coefficients of the mode 2 and 3 show a decreasing trend, and both are positive in 2016 and negative in 2019."

[Figure]

Figures 12. Distribution of eigenvectors contribution rate (a-c) and time coefficients (d-f) for the first

three modes of SD.

*14. Please discuss the simulation bias in SD dataset. Please also discuss the limitation of the current SD dataset.*

Thanks to your suggestion, the discussion on modeling bias has been added to the "5. Discussion" based on the answers to questions 7 and 8, and the limitation of the current SD dataset has been also added to the "5. Discussion" based on the answers to two main questions **in the revised manuscript.**

---

## Author Comment (AC2)

**Dear Reviewer 2, Thank you very much for your two valuable questions, we will answer them and add the responses to the Introduction and Discussion sections of our paper.**

*Sunshine data is generally used to estimate global radiation, rarely the other way round, so the purpose and significance of this paper is unclear.*

We apologize for our lack of clarity in the introduction.

Sunshine duration (SD) the is widely used for Global Radiation (GR) estimation and was presented in FAO56 (Allen et al., 1998), and subsequently many relationship models with GR have been proposed for the SD to the maximum SD available (N) ratio (Chen et al., 2019; Prieto et al., 2022). SD measurement is long, continuous, spatially dense, and reliable, and are considered to be the best alternative to solar radiation (Xia et al., 2010). SD is an important indicator for assessing climate change trends, the probability of human diseases (Chang et al., 2022; Gu et al., 2019) and a crucial parameter for seasonal carbon cycle modeling (Fang et al., 2022; Zhao et al., 2021).

The two main ways of estimating GR, satellite remote sensing and ground measurement, are presently biased. The satellite remote sensing may be **susceptible to atmospheric inverse radiation from clouds and aerosols** (Mei et al., 2018; Letu et al., 2020) for its sensors receive reflectance information from land surface. **The GR ground measurement stations in the Chinese region are very few (145), much fewer than conventional meteorological stations (2408) which observed SD**, even on a global basis, the distribution of radiation stations is very sparse (Bao et al., 2019; Wang et al., 2012). Therefore, the large-area GR verification is difficult.

Although SD is often used to estimate GR, it may be necessary to utilize high-precision SD as a support for obtaining higher precision GR, so we believe that it is necessary to invert high spatiotemporal precision SD against the background of the low spatial and temporal resolution and reproduction errors in the GR remote sensing data.

Reference:

Allen, R.G., Pereira, L.S., Raes, D., & Smith, M. (1998). Crop evapotranspiration: guidelines for computing crop water requirements.

Bao, S., Letu, H., Zhao, J., Lei, Y., Zhao, C., Li, J., Tana, G., Liu, C., Guo, E., Zhang, J., He, J., & Bao, Y. (2019). Spatiotemporal distributions of cloud radiative forcing and response to cloud parameters over the Mongolian Plateau during 2003–2017. International Journal of Climatology, 40, 4082 - 4101.

Chang, Z., Chen, Y., Zhao, Y., Fu, J., Liu, Y., Tang, S., Han, Y., & Fan, Z. (2022). Association of sunshine duration with acute myocardial infarction hospital admissions in Beijing, China: A time-series analysis within-summer. The Science of the total environment, 154528.

Chen, J., He, L., Yang, H., Ma, M., Chen, Q., Wu, S., & Xiao, Z. (2019). Empirical models for estimating monthly global solar radiation: A most comprehensive review and comparative case study in China. Renewable and Sustainable Energy Reviews.

Fang, J., Shugart, H.H., Liu, F., Yan, X., Song, Y.X., & Lv, F. (2022). FORCCHN V2.0: an

individual-based model for predicting multiscale forest carbon dynamics. Geoscientific Model Development.

Gu, S., Huang, R., Yang, J., Sun, S., Xu, Y., Zhang, R., Wang, Y., Lu, B., He, T., Wang, A., Bian, G., & Wang, Q. (2019). Exposure-lag-response association between sunlight and schizophrenia in Ningbo, China. Environmental pollution, 247, 285-292.

Letu, H., Yang, K., Nakajima, T.Y., Ishimoto, H., Nagao, T.M., Riedi, J.C., Baran, A.J., Ma, R., Wang, T., Shang, H., Khatri, P., Chen, L., Shi, C., & Shi, J. (2020). High-resolution retrieval of cloud microphysical properties and surface solar radiation using Himawari-8/AHI next-generation geostationary satellite. Remote Sensing of Environment, 239, 111583.

Mei, L., Rozanov, V.V., Vountas, M., & Burrows, J.P. (2018). The retrieval of ice cloud parameters from multi-spectral satellite observations of reflectance using a modified XBAER algorithm. Remote Sensing of Environment.

Prieto, J. I., & García, D. (2022). Global solar radiation models: A critical review from the point of view of homogeneity and case study. Renewable and Sustainable Energy Reviews, 155, 111856.

Wang, T., Yan, G., & Chen, L. (2012). Consistent retrieval methods to estimate land surface shortwave and longwave radiative flux components under clear-sky conditions. Remote Sensing of Environment, 124, 61-71.

Xia, X. (2010). Spatiotemporal changes in sunshine duration and cloud amount as well as their relationship in China during 1954-2005. Journal of Geophysical Research, 115.

Zhao, J., Liu, D., Cao, Y., Zhang, L., Peng, H., Wang, K., Xie, H., & Wang, C. (2021). An integrated remote sensing and model approach for assessing forest carbon fluxes in China. The Science of the total environment, 152480.

*Secondly, the author mentions the shortcomings of the Himawari-8 official radiation data in the introduction, so it is confusing to use such a flawed data as the basis for the SD estimation.*

We apologize for the confusion of the article. **We have described the advantages of this AHI in terms of spatiotemporal resolution** before presenting the shortcomings of the its radiation data (this part of the introduction will be explained in more detail in the revised manuscript). The problem of the Himawari AHI radiation data is not an isolated case, but is a problem that exists in all satellite remote sensing radiation data (as explained in the first question). Meanwhile, we had added the advantages of AHI over other geostationary satellites in the "Data and method" section in the new revised version of the manuscript.

---

## Author Response (AR2)

**Dear Reviewer 1, After receiving your comments, we immediately prepared a response, and the following is a point-by-point response to all the questions.**

*1. I still think the importance of daily sunshine duration data is not highlighted enough in introduction. Besides used in A-P model for estimating global solar radiation, could you give us some sepecific example on the application of sunshine duration?*

Thanks for this valuable question. SD is a key parameter of solar power potential forecasting, climate change assessment and agricultural production, and some researchers have found that changes in SD also affect the probability of human diseases. We have added specific examples for each of the above three SD application scenarios. (See revised manuscript Line 43-53)

*2. Line 64 in tracked changes: If there are bias in satellite retrived solar radiation, why did you invert sunshine duration from Himawari AHI?*

Thank you for your comment.

Remote sensing solar radiation products do have errors due to estimation based on ground reflection information, which is unavoidable.

The Himawari AHI radiation product has a leading edge in terms of temporal resolution for daily-scale radiation estimates, but its errors need to be corrected for by ground-based observation sites. However, the limited number of radiation observation stations and the lack of empirical physical models for satellite-ground radiation correction in China and other parts of the world constrain the accuracy of satellite remote sensing radiation products.

SD is a readily available and cost-effective indicator for monitoring the global radiation resources, there are more than 2,000 regular meteorological stations observing SD in China, which is much higher than the radiation observation stations (145). And there is also a widely used empirical physical model between SD and solar radiation (A-P model). The above two points provide strong support for the calibration of the Himawari AHI radiation products, and we can calibrate the AHI radiation data to high-resolution grid-point SD data based on a large number of SD ground observation stations.

Generally speaking, we believe that the study on SD estimation is easy to validate, and the SD data are more credible theoretically. Until the radiation observation system in China is well developed, we believe that the study on the estimation of SD is valuable.
We have made major changes to the "introduction" section to make it as logical as possible.

*3. There is a widespread transition from manual to automatic sunshine duration recorders in 2019 or station relocations. Did you consider this effect on validation by ground observations? Please refer to the paper https://essd.copernicus.org/preprints/essd-2024-493/*

Thank you for your such this valuable question, we have taken this into account and therefore used both 2016 and 2023 as validation sets, and we have added additions to the review and cited the literature in the revised manuscript (see Line 136-137).

*4. Besides AOD and water vapor, could you analyze the effect of cloud cover on SD estimation?*

Thank you for your comment, we have added the analysis the effect of cloud cover on SD estimation. (See revised manuscript Line 234-237 and Figure 10,11)

*5. Why there is no data for regions 80ºE westward in figure 10, 11 and 12? I do think there are SD observations for 80ºE westward in Figure 1.*

We apologize for the confusion, Himawari AHI's coverage is at 80ºE eastwards (mentioned in revised manuscript Line 106-107), but we still believe that AHI data is not alternative in our research and for the high temporal and spatial resolution and long acquisition time period.

**Dear Reviewer 2, We apologize that the introduction to our manuscript, especially the introductory section, did not enable you as well as the Reviewer 1 to fully understand the significance of the study. We have made major changes to the "introduction" section to make it as logical as possible.**

Satellite remote sensing is an effective method of monitoring and tracking solar radiation (mainly known as GR). However, GR inversion by satellite sensors based on reflectance information from the land surface is highly susceptible to atmospheric inverted radiation from clouds and aerosols, which need to be corrected for by ground measurement radiation stations.

However, the limited number of radiation observation stations and the lack of empirical physical models for satellite-ground radiation correction in China and other parts of the world constrain the accuracy of satellite remote sensing radiation products.

SD is a readily available and cost-effective indicator for monitoring the global radiation resources, there are more than 2,000 regular meteorological stations observing SD in China, which is much higher than the radiation observation stations (145). And there is also a widely used empirical physical model between SD and solar radiation (A-P model). The above two points provide strong support for our study.

We believe that the study on SD estimation is easy to validate, and the SD data are more credible theoretically. Until the radiation observation system in China is well developed, we believe that the study on the estimation of SD is valuable.

(**We have described it more fully in the "introduction". We would like to thank you and look forward to your consideration!**)